# Chemical unclonable functions based on operable random DNA pools

Anne M. Luescher [1], Andreas L. Gimpel [1], Wendelin J. Stark[1], Reinhard Heckel [2] & Robert N. Grass [1] ✉

Physical unclonable functions (PUFs) based on unique tokens generated by random manufacturing processes have been proposed as an alternative to mathematical one-way algorithms. However, these tokens are not distributable, which is a disadvantage for decentralized applications. Finding unclonable, yet distributable functions would help bridge this gap and expand the applications of object-bound cryptography. Here we show that large random DNA pools with a segmented structure of alternating constant and randomly generated portions are able to calculate distinct outputs from millions of inputs in a specific and reproducible manner, in analogy to physical unclonable functions. Our experimental data with pools comprising up to >$10^{10}$ unique sequences and encompassing >750 comparisons of resulting outputs demonstrate that the proposed chemical unclonable function (CUF) system is robust, distributable, and scalable. Based on this proof of concept, CUF-based anti-counterfeiting systems, non-fungible objects and decentralized multi-user authentication are conceivable.

Non-biological applications of DNA have gained importance due to the unique chemical properties of nucleic acids[1]. DNA's extraordinary information density of 455 exabytes per gram[2] and the available molecular writing, reading and editing technologies extend its applications far beyond genetics. Notably, synthetic DNA is already being used for digital data storage[2–4], barcoding[5] and steganography[6]. In addition, DNA computation has emerged as an interdisciplinary field that makes use of the available biomolecular tools to perform calculations[7]. Nucleic acids have since been successfully used to solve combinatorial problems[8] as well as computationally hard tasks[9], and were implemented in logic gates[10] and for random number generation[11]. Even DNA-based programmable gate arrays for general purpose computing have been introduced[12], as well as programmable automata[13]. As the cost for chemical synthesis and sequencing of DNA have dropped dramatically with the advent of the twenty-first century[14], research in DNA-based information technology has opened up toward even more advanced applications.

In parallel to these developments in DNA research, digital transformation has led to the routine use of cryptography in applications related to authentication and encryption, electronic access control and digital payment[15]. An important cryptographic tool are one-way functions, which calculate an output value from an input using mathematical operations that are relatively easy to perform in one direction, but computationally infeasible to invert[16,17].

Although mathematical one-way functions are widely used, advancements in quantum computing and the lack of proof for cryptographic security of such algorithms have led to the exploration of alternative methods.[17,18] For example, Pappu et al.[18] suggested the use of an object with a disordered microstructure for cryptographic key generation. They exploited the randomness of silica spheres suspended in a hardened epoxy to map the orientation of the token in relation to a light source (input) to the resulting laser scattering pattern (output). Such functions work similar to cryptographic hash functions, but rely on a physical source of disorder instead of number theory, and have generally been termed physical unclonable functions (PUF)[19]. PUFs are characterized by their ability to translate an input (challenge) to an output (response) through a physical system that is unique and cannot be replicated, with the challenge response pairs

[1]Department of Chemistry and Applied Biosciences, ETH Zürich, Vladimir-Prelog-Weg 1-5, 8093 Zürich, Switzerland. [2]Department of Computer Engineering, Technical University of Munich, Arcisstrasse 21, 80333 Munich, Germany. ✉e-mail: robert.grass@chem.ethz.ch

**Table 1 | Comparative features of one-way functions, PUFs and CUFs**

| | One-way function[17] | PUF[18] | CUF (this work) |
|---|---|---|---|
| Manifestation | Abstract, mathematical concept | Physical/tangible object(s) | Large pool of molecules |
| Function generation | Not generated; discovered as mathematical function | Random manufacture of a unique token | Random DNA synthesis of a unique pool |
| Operation/execution | Digital | Physical | Chemical |
| Readout | Exact number | Noisy data[a] | Noisy data[a] |
| Clonability | Yes | No | Copiable and uncopiable state |
| Consumed by execution | No | No | Yes |
| Distributability | Yes | No | Yes |

[a]Numeric signature key can be extracted from the data.

(CRPs) being difficult or impossible to predict. PUFs have been proposed for applications in intellectual property protection[20], public key cryptography[20] and anti-counterfeiting of goods and services[21,22].

Genetic information has already been suggested as a medium for physical unclonable functions by using CRISPR-induced non-homologous end joining repair to generate a unique barcode-indel mapping (CRISPR-PUFs).[23] There, the random process refers to the combination of barcodes and indels in a given cell line. We instead propose to directly use randomly generated DNA sequences, giving rise to massive levels of entropy. As it was recently shown that chemical DNA synthesis can be used to generate random numbers[11], we envisioned that enormous random DNA pools could be used to implement a type of object-bound cryptography based on chemistry instead of physics. In this work, we introduce chemical unclonable functions (CUFs) capable of performing calculations by controlled molecular operations. We show that such CUFs are robust, scalable and secure, and that their properties can be compared with PUFs and one-way functions (Table 1). Furthermore, and in contrast to physical functions, the implemented system allows for switching between a copiable and an uncopiable state. Based on our results, we suggest use cases of decentralized multi-user authentication and non-fungible items, connecting the digital with the physical world.

## Results

### Chemical unclonable functions

To implement a CUF, a large pool of random DNA is used as a source of entropy. Random information in the form of DNA can be obtained through a random chemical synthesis approach (Fig. 1b and Supplementary Note 1). This process uses a mix of the four nucleobases for synthesis, with the entropy of the mix leading to random incorporation at any given position in a growing DNA strand. Such an approach has the advantage of combinatorial parallelization, enabling the generation of trillions of unique strands of unknown sequence at low cost (Supplementary Note 2). However, despite this, specific base combinations are addressable by PCR. The chemical specificity of Watson-Crick-Base pairing dictates that, in theory, even in a DNA pool of unknown composition, a set of PCR primers will favorably bind to, and thus select, the complementary sequence to pair with. The action of the polymerase subsequently leads to an exponential copying of the section located between the primers[24]. As the amplified sequences originate from random chemical synthesis, the resulting amplicons are unknowable. This process as illustrated in Fig. 1d corresponds to providing the system with an input (a binary number mapped to a set of primer sequences) to yield a reproducible, but unpredictable, chemical readout (the sequences amplified in the reaction). The readout is identified through sequencing and mapped to a numeric output.

### Design of operable random DNA (orDNA) libraries

Despite these properties, purely random DNA is impractical to implement a useful chemical unclonable function. In order to be used for cryptographic applications in analogy to PUFs or one-way functions, the same challenge-response pair must be generated at least twice (once for registration, and at least a second time for authentication). However, in a purely random DNA pool of significant entropy, every sequence in the pool is expected to be unique (Supplementary Note 3). Due to this, and because PCR inherently modifies the composition of the DNA template pool, any outcome can only be produced once. Moreover, selecting sequences from a purely random pool would make readout and data evaluation challenging, as these processes work best in the presence of identifiable portions and constant readout lengths. We addressed this by designing a library containing sequence-determined parts in addition to randomly synthesized segments. The added constant portions allow for copying of all the sequences in a synthesized pool, independent of the composition of their random segments. In addition, we further structured the library such that functional differentiation between the random segments becomes possible. This idea of operable random DNA (orDNA) with addressable constant and random segments ensures that the randomness generated by the synthetic process is experimentally accessible and uniformly structured.

The detailed design of such a functional orDNA library is sketched in Fig. 1a and fully represented in Supplementary Fig. 1. An individual sequence consists of three separate random segments, each of which is flanked by constant regions. Two of the random parts are intended to bind the input primers. A third random segment is located between the two input regions and is amplified if the primers successfully bind to the input regions located up- and downstream of a given strand. Two constant segments separating the output from the input regions are sequencing adapters used for effortless readout of the output. Two handle segments, located at either sequence end, allow for PCR amplification of the entire DNA pool to generate the copies needed for multiple operations. Such a DNA pool works as a chemical function, i.e., it can be operated to generate challenge-response pairs via random input/output combinations present in the pool. While it cannot be re-generated from scratch, it can be copied at will using the handles by any actor who has physical access to the pool and detailed knowledge of the handle segments. Cryptographic security is thus only guaranteed as long as no malignant actor gains access to the pool and the knowledge of its structure. In order generate a more secure unclonable function, the copiable pool can optionally be transferred to a second, unclonable state, in which the function can still be operated on, but further copying is disabled. To implement this, we have generated a second random DNA library (Supplementary Fig. 2), which allows the irreversible removal of the amplification handles via restriction digest and sub-sequent introduction of 2',3'-dideoxy nucleotide analogs at the 3'-end, thus permanently disabling the generation of exact copies of the entire pool. Consequently, the combination between the pool's intrinsic randomness and the subsequent chemical edits guarantees the unclonability of the function. Re-creating the same pool from scratch would require knowledge of the entire composition as generated by the random manufacturing

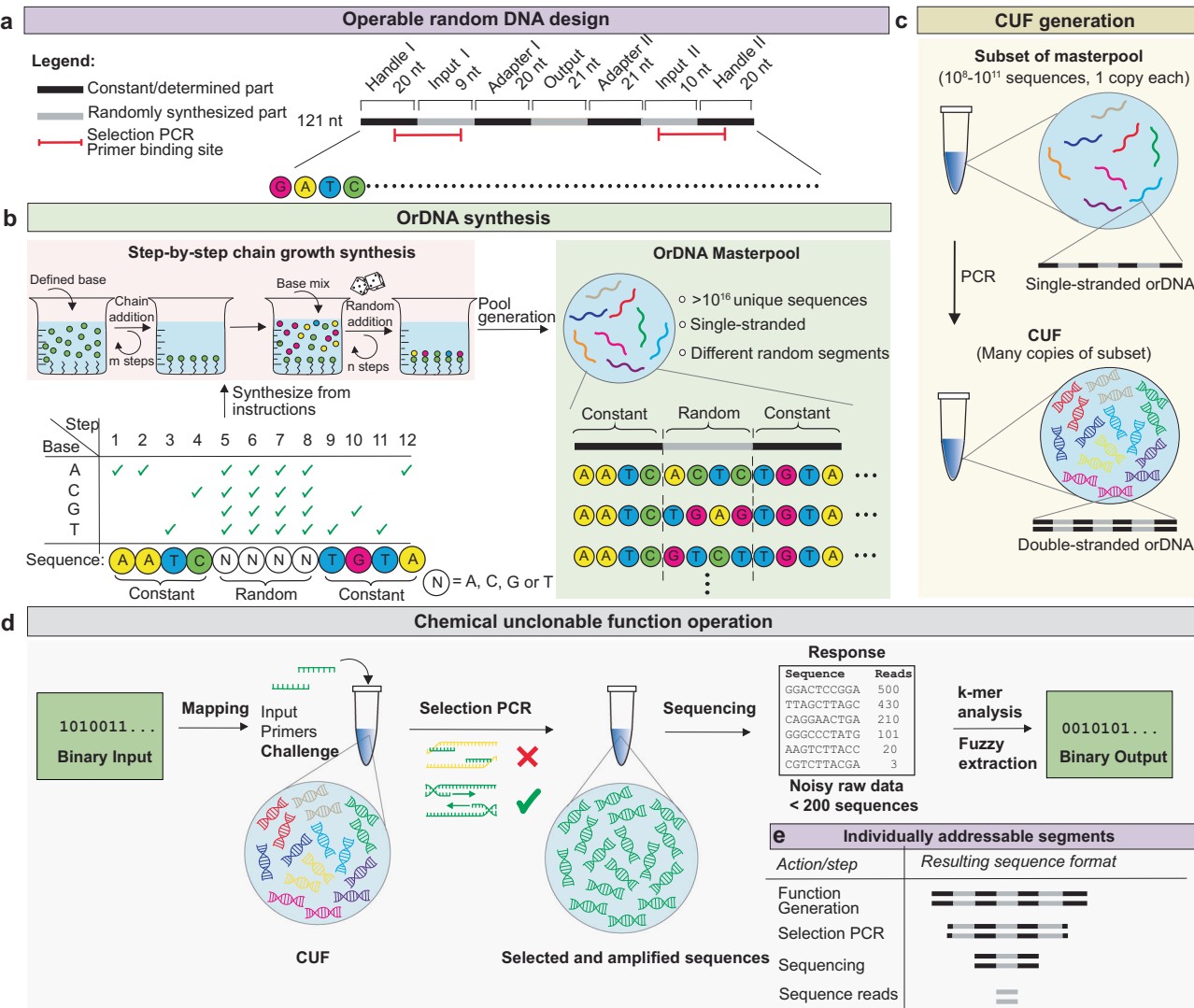

**Fig. 1 | Design and working principle of DNA-based chemical unclonable functions. a** Library design of the implemented operable random DNA (orDNA), containing constant handle and adapter segments that separate three random elements (two input and one output segment). The segment in red indicates the binding site of the input primers. **b** Step-by-step chemical synthesis of orDNA, combining random with sequence-determined synthesis, as offered by many commercial suppliers. **c** CUF generation procedure, amplifying a subset from the synthesized orDNA masterpool to generate multiple double-stranded copies of each sequence. **d** Working principle of a DNA-based one-way function. A numeric input is mapped to two primer sequences, which selectively amplify matching sequences out of a large DNA pool of a random composition. The resulting selection is sequenced, leading to a "fingerprint" characteristic for any given input, from which a binary, fixed-length output is generated using numeric data processing (k-mer analysis and fuzzy extraction). **e** Overview of the different orDNA segments involved in each step from function generation to readout.

process, followed by controlled re-synthesis of all unique sequences. Both are prohibitively time-consuming and costly due to the large pool size, and additionally rendered difficult to impossible by the truncated ends that prohibit global PCR and ligation.

## Practical implementation of the orDNA function

DNA libraries are accessible through chemical synthesis on solid support and can be purchased from commercial suppliers. In sequence-determined synthesis, each synthetic step adds a single base to the many growing chains in a controlled manner. At positions where a random base is desired instead, an equimolar mix of the four bases is used for the respective synthetic cycle[11], with the entropy inherent to the mix leading to a random base incorporation, as illustrated in Fig. 1b). The two synthesis modes can be combined in a protocol to yield the desired combination of determined and random segments within a synthesized sequence. As there are many random chains within a parallelized synthesis batch, this method leads to a library with

up to $4^n$ different sequences, where $n$ is the number of randomly synthesized positions in each sequence.

The implemented library contains a total of 40 random nucleotides per sequence, meaning a potential set of $4^{40} \approx$ ca $10^{24}$ random sequences. Out of this vast space of combinatorial possibilities, a typical synthesis yields ~$6 \cdot 10^{16}$ unique single-stranded sequences. When calculating with the theoretical density of 2 bits per base, such a masterpool contains ~$5 \cdot 10^{18}$ bits (0.6 exabyte) of chemical entropy, at a cost of <100 USD. In contrast, synthesizing the same number of determined sequences would be infeasible. Companies specializing in synthesis of large DNA pools currently offer prices of ~50,000 USD for a batch of a million oligos, of which several billion batches would be required to amount to $6 \cdot 10^{16}$ sequences. This asymmetry in time and cost means that no such masterpool can be re-created with intent.

To generate double-stranded orDNA from this single-stranded masterpool, a subset of the sequences was copied by running a PCR with the outer handle sequences as primers (Fig. 1c). This PCR yielded a

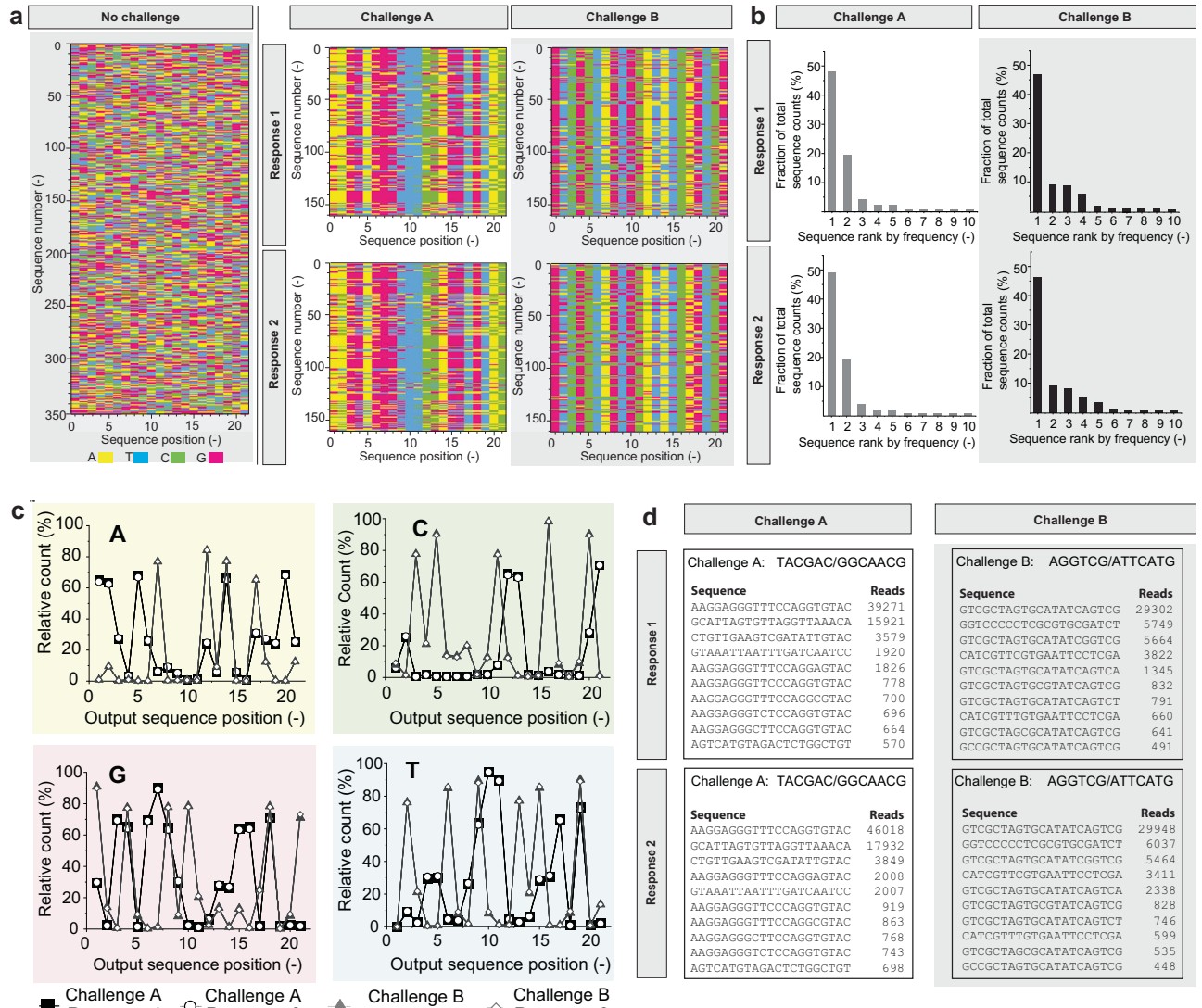

**Fig. 2 | Multi-level challenge-response-pair analysis. a** Positional base content (A, C, G, T) of arbitrary subsets of sequence reads generated as a response from separate challenges with experimental duplicates (response 1 and response 2). As a comparison, the equivalent result generated by non-selectively reading random DNA is shown, as found in Meiser et al.[11]. The *x*-axis represents the 21-mer composition of the orDNA's output segment, the *y*-axis the (arbitrary) sequence number within the analyzed set. **b** Relative counts of the 10 most frequent output sequences, for the same challenge response pairs (CRPs) as in (**a**). **c** Relative frequency of A, C, G and T across the 21 positions of the output across the entire read set resulting from Illumina sequencing of the same CRPs as in (**a**). **d** Examples of datasets as used for further numeric processing, showing the ten most frequent sequences with their absolute read count resulting from the same CRPs as in (**a**). Source data are provided as a Source Data file.

mean of ca 16,000 copies per sampled sequence. The desired number of sequences (varying from ~$10^8$–$10^{10}$ in this work), were picked from the masterpool by pipetting the corresponding volume from an aqueous solution of known DNA concentration. With this procedure, the scale of the CUF and thus the number of supported challenge-response pairs can be controlled. This sub-selection step represents an additional random process (urn sampling without replacement), ensuring that not even an actor having the masterpool can re-create a given subpool. In consequence, CUFs also fulfill manufacturer resistance[25].

**Operation of the orDNA to generate challenge-response pairs**
The first set of experiments were conducted on CUF S1, comprising $10^8$ unique sequences (according to design in Supplementary Fig. 1) containing an equivalent of 1 GB of random information, with the aim of assessing whether input-output pairs can be reproducibly generated. PCR was conducted using different sets of input primers (the

"challenge" to the system), which were designed to bind to both the input regions and a part of the constant regions (Supplementary Note 4). Even though nothing is known about the random sequence composition, this PCR exponentially amplifies the DNA strands that contain input segments matching the primers above the large background of non-matching input portions. The resulting amplicons are segmentally ordered and of constant length, enabling collective readout by next-generation sequencing (NGS).

The response to any challenge to the CUF essentially consists of a set of 21-mers, corresponding to the output segments of the read sequences. Figure 2a shows arbitrary subsets of such sequencing readouts with color-coded bases. Even before quantitative analysis, comparison of two different challenges (each conducted twice) on the raw data level (after filtering for presence of expected constant regions), visualizing base content by position shows that two responses belonging to the same input/challenge are qualitatively similar, and

responses to a different challenge lead to a different set of amplicons. All responses, although noisy, have a visibly reduced randomness as compared to a readout of purely random sequences generated without a selection step. Analysis of the read frequency distributions of the most read sequences (Fig. 2b) and of base counts by position over the entire respective read sets (Fig. 2c) further show this observable reduction of sequence diversity and the qualitative reproducibility of CUF readout.

## Set similarity calculation

To put these qualitative observations into quantitative terms, the sequence information in the form of sequence reads and frequency, as exemplified in Fig. 2d, needs to be translated into a metric in order to assign two responses to either the same or different challenges. This metric needs to account for noise and potential errors, as two responses to the same challenge can slightly vary. This can be explained by the fact that synthesis, PCR, sequencing, and pool sampling are inherently noisy processes. Specifically, off-target amplification can occur, in particular when the template concentration is low relative to the background[26]. Furthermore, PCR has a non-negligible mutation rate of up to $2 \cdot 10^{-4}$ per base[27], resulting in a distribution of sequence variants in the output. In addition, and due to the limited copy number of unique sequences in the pool, stochastic effects arise in that some sequences are drawn more often than others. In consequence, while the distribution of sequences of two samples taken from the same pool are similar, they are typically not exactly the same, resulting in a slightly different readout.

To resolve this issue, we applied a signature extraction based on set similarity. This is permissible, as the output space (a set of random 21-mers) is significantly larger than the input space (13 random nts). To this end, after filtering the data, we used a k-mer extraction routine to computationally compare the sequence sets, which is a tool commonly used in genomic data analysis[28] and beyond. The similarity of the extracted k-mer sets was then quantified with a weighted Jaccard coefficient.

This procedure outputs a score between 0 and 1 for any two compared sets, with 1 corresponding to fully identical sets. Supplementary Notes 6–8 provide a more detailed discussion of the methods applied and the selection of the parameters.

## Experimental similarity assessment

We applied this procedure to a wide range of experiments, comparing different inputs and outputs under various conditions. Figure 3a shows the output similarities between the responses generated from different input challenges. Comparison with the truth matrix shows that the measured similarities correspond to the expected outcome: Like challenges have highly similar responses and unlike challenges lead to dissimilar responses. We started with running a randomly chosen challenge (C1) through CUF S1 multiple times. While the input sequence was chosen at random, the number of input bases within this initial primer pair was chosen such that given the pool size, the expectation value of the frequency of perfectly matching sequences in the pool was ~1.5 (see Supplementary Note 4). As expected, the responses show consistently high similarities with scores between 0.8 and 1 (experiments 1–4 in Fig. 3a). As it would be infeasible to test all possible CRPs, we evaluated the CUF's robustness by focusing on some of the most difficult scenarios, i.e., comparing challenges with the smallest possible variations (Levenshtein distance $D_L = 1$) to each other (experiments 5–8, 13–16). The results show that if a single base in the input primer is changed, the CUF returns a completely unrelated response. Consequently, the number of challenge response pairs is $4^n$, $n$ being the number of input bases. If the input is set to 13 bases (experiments 1–12), this results in ~67 million CRPs. Further tested demanding challenges even included poly-T repeats and variable-

length inputs. We also generated two more pools (CUF S2 and S3) by using new subsets—again comprising $10^8$ sequences—from the masterpool. As these sequences are unrelated to the previously used CUF S1, when run with challenge C1, the new CUFs should return entirely different responses to any given challenge. This was experimentally confirmed (experiments 20–25, Fig. 3a). Combined, these results suggest that there are no apparent constraints in the allowed input space beyond the fact that the physical length of the challenge (in bp) is limited by the primer length and the diversity of the pool comprising the function (Supplementary Note 4).

## Enabling multiple operations and distributability

In addition to returning reliable outputs, an important criterion for any practically useful function is that it can be evaluated multiple times. In contrast to mathematical functions, this is not a given for chemical functions. Each evaluation of the function uses up a part of the pool that is altered in its chemical composition by the process and, therefore, irretrievable in its original form. To ensure that the function can be evaluated reliably many times nonetheless, we amplified the entire pool of CUF S1 in five proliferations (creating daughter generations P1–P5) using the outer handles as primers. Each amplification step uses the product of the previous one as template (Fig. 3b). Figure 3c shows the consistency of the output throughout the daughter generations, whereby the same randomly chosen base challenge (C1) was used as an input as for the initial replicates (further challenge-response pairs are shown in Supplementary Fig. 4). The resulting outputs still showed high similarities across all generations, as opposed to e.g., comparing outputs resulting from inputs differing by a single base. While any PCR bias introduced throughout the proliferation steps cannot be quantified due to the pools being too large to be sequenced within reasonable time and cost constraints, the robustness in terms of differentiating input-output similarity is maintained across the generations. Based on the observation that on average the amount corresponding to ~200 function operations was generated of each daughter generation, this leads to more than $6 \cdot 10^{13}$ possible evaluations, with no indication that this is near to the theoretical or practical limit. This ability to be copied while retaining operability makes a CUF distributable to multiple users. Each user can thus get access to one or multiple identical copies, which can be viewed as tokens representing the same function.

## Scaling-up the function and increasing the address space

A further property of interest is the scalability of chemical one-way functions to increase the number of possible inputs and outputs. Two larger CUFs, M1 and L1, were therefore subsequently generated using ~$1.6 \cdot 10^9$ and $2.6 \cdot 10^{10}$ unique sequences from the masterpool, respectively (as opposed to $10^8$ sequences for CUFs S1–S3). Again, inputs differing by a Levenshtein distance of 1 were compared with the responses matching the expected outcome, showing that PCR enrichment still works reliably over an overwhelming amount of background sequences (Fig. 3d).

## Numeric key generation

While response similarity provides a good qualitative measure of the CUF's functionality, for some digital applications it is necessary to map the noisy responses into an unambiguous numeric key (Fig. 3h). To achieve this, we use a MinHash algorithm followed by a fuzzy extractor (Supplementary Note 10). MinHash-based tools are widely used, with numerous applications in genomics, such as taxonomic diversity assessment[29] and metagenome distance estimation[30]. The applied algorithm maps individual sets—in this case the k-mers that comprise the responses of a CUF—to signature vectors, which are then compared in terms of their Hamming distance ($D_H$). In a perfect dataset without noise, $D_H$ for any two MinHashes generated by like

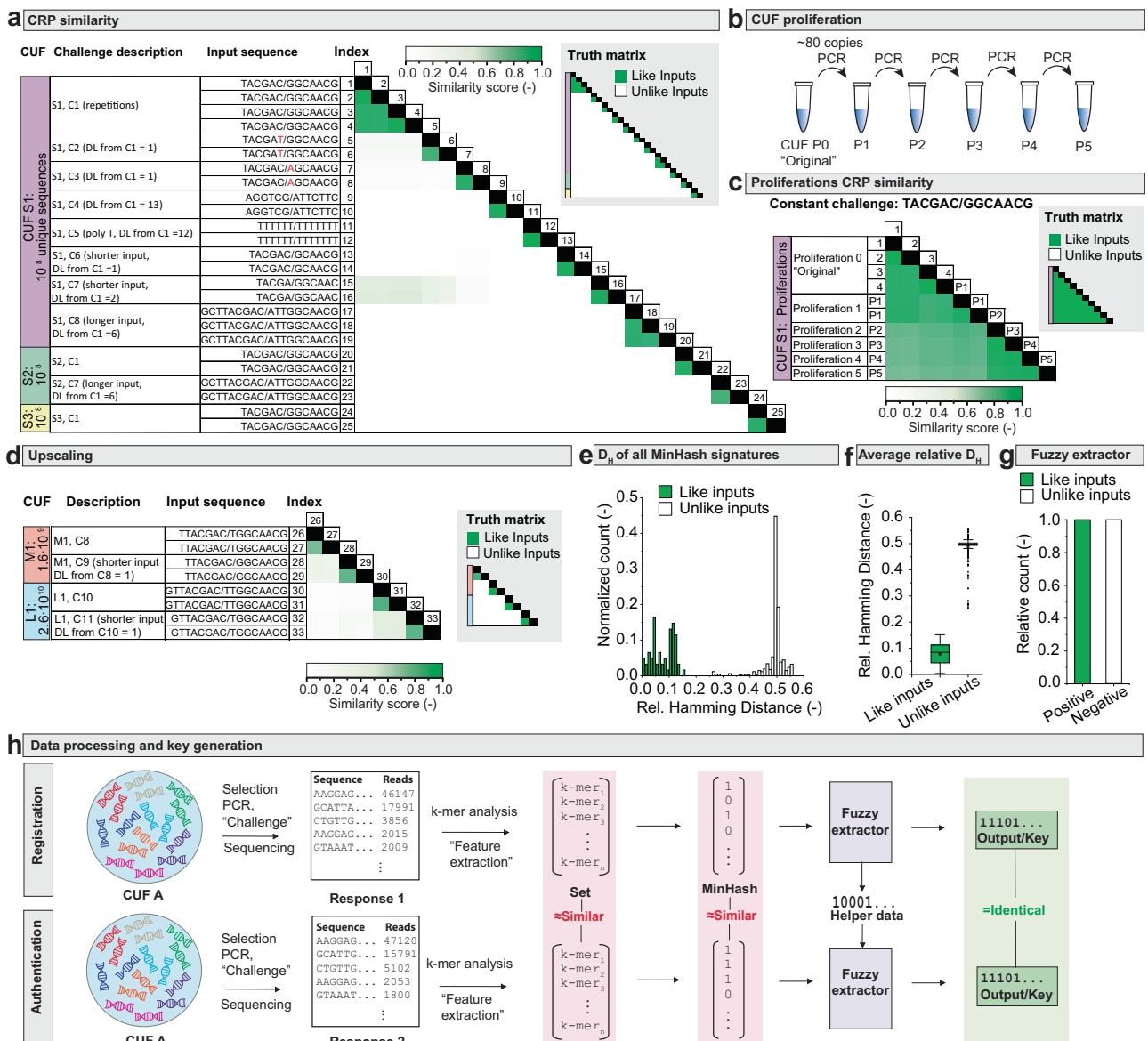

**Fig. 3 | CUF evaluation and key generation. a** Correlation matrix comparing experimentally measured similarity scores (in green color gradient) as calculated from k-mer sets extracted from the sequencing reads for all experiments performed on chemical unclonable functions (CUFs) S1, S2 and S3, comprising $10^8$ unique sequences each (Supplementary Note 3). Indices correspond to the experiment number. Notable relationships between the different challenges are described next to the input sequences, including Levenshtein distances $D_L$. The truth matrix indicates which correlated indices stem from like and unlike inputs. **b** CUF proliferation method to produce daughter generations (P1–P5) of the same CUF by PCR. **c** Correlation matrix and truth table in analogy to (**a**), comparing five proliferations of CUF S1 in their respective response to the same challenge. **d** Correlation matrix and truth matrix in analogy to (**a**), comparing the larger CUFs M1 and L1 in the edge case of a minimal Levenshtein distance ($D_L$) between the

challenges. **e** Histogram of relative Hamming distances between MinHashes of cross-comparisons between the 39 measured challenge response pairs (CRPs) as shown in (**a**)–(**d**). $n = 680$ comparisons between unlike inputs, $n = 61$ comparisons between like inputs. **f** Boxplot showing average relative Hamming distances, as per the distributions in (**e**). $n = 680$ comparisons between unlike inputs, $n = 61$ comparisons between like inputs. Indicated are the median (middle line), mean (circled dot), 25th and 75th percentile (box) and 1.5 interquartile range (whiskers), with outliers marked as black dots. **g** Assignment of compared responses to belonging to same (positive) or different (negative) challenges by the fuzzy extractor. $n = 680$ comparisons between unlike inputs, $n = 61$ comparisons between like inputs. **h** Schematic of CRP generation and data processing to generate and compare numerical keys from noisy sequencing data, involving k-mer analysis, MinHash generation and fuzzy extraction. Source data are provided as a Source Data file.

inputs is zero, while the expected average relative distance of Min-Hashes stemming from unlike inputs is 0.5—i.e., essentially random. This closely matches the experimental results, which show two well separated distributions (Fig. 3e) with an average Hamming distance of 0.1 for like inputs and 0.5 for unlike inputs (Fig. 3f). To fully eliminate the noise, the MinHashes are then fed into a fuzzy extractor algorithm, which generates a 256-bit key and a string of public helper data for error correction. In the dataset of 39 experiments as shown in Fig. 3, including minimally different challenges as

described above, the fuzzy extractor was 100% accurate in correctly distinguishing keys derived from like or unlike inputs (Fig. 3g). A discussion of the robustness of the parameter choice can be found in Supplementary Note 7. Thus, the implemented system transforms noisy sequencing data into an unambiguous output that is constant and expected to be unique for any challenge-response-pair generated by a CUF, and which reliably determines if two outputs stem from the same or from different inputs. This is comparable to the output of a mathematical one-way function.

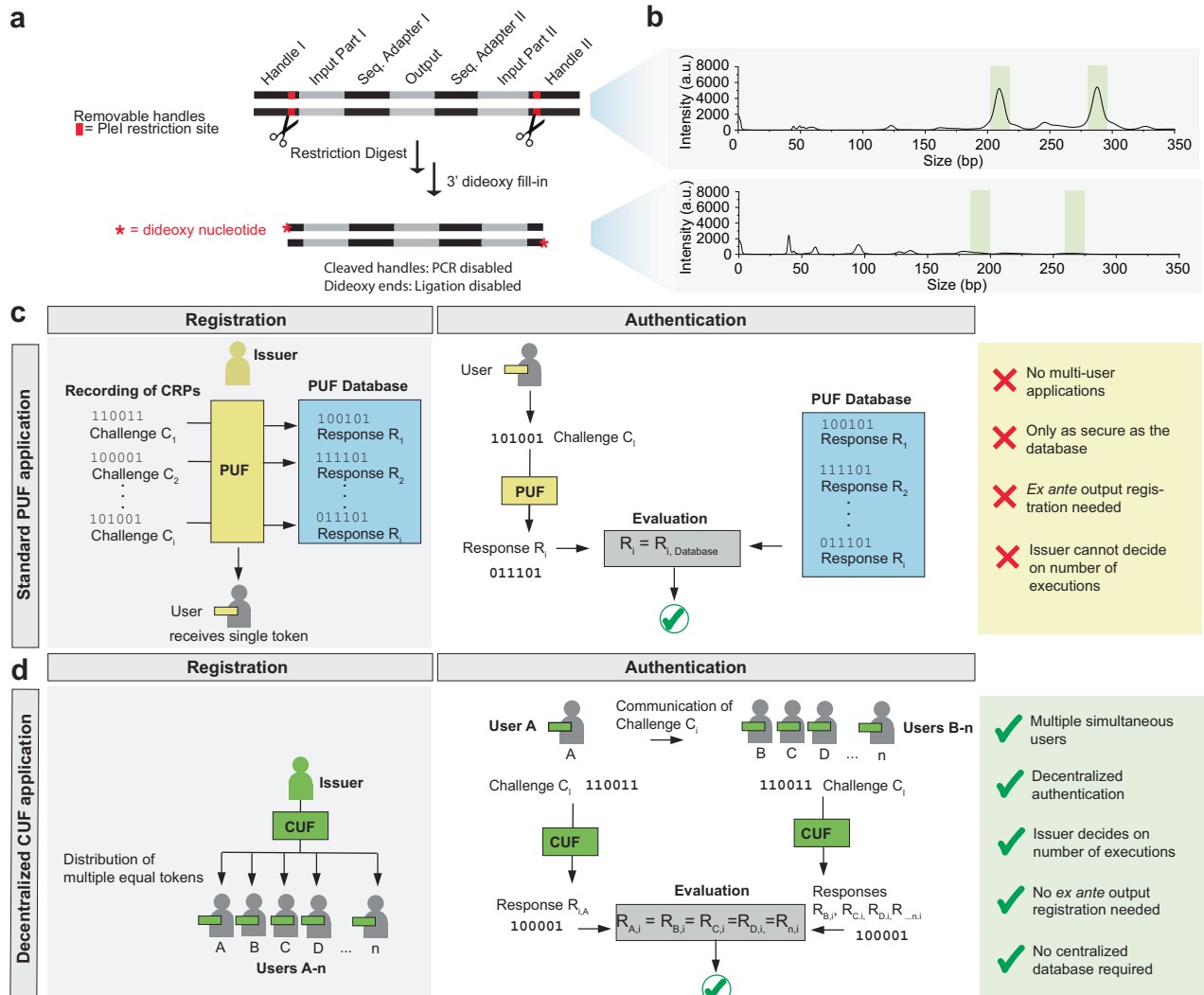

**Fig. 4 | Application of CUFs in decentralized multi-user systems. a** Procedure to change a chemical unclonable function (CUF) from a copiable to an uncopiable state. **b** Electropherograms of the ligation products of a CUF in the copiable and uncopiable state. The green areas show the approximate expected length of the single-end and paired-end ligated products, respectively. **c** Exemplary application scenario for a PUF. Upon manufacturing, a list of challenge response pairs (CRPs) for the PUF token is recorded. For evaluation of a request, the CRP generated by the PUF is compared to the database. Adapted from McGrath et al.[35]. **d** Extended application scenario for a CUF. The issuer generates and distributes a pre-determined amount of CUF tokens to multiple users. Any user in possession of a token belonging to a given CUF can then start an authentication request, while the others are able to confirm the user's authenticity in a decentralized manner. Source data are provided as a Source Data file.

## Introducing irreversibility and unclonability to the CUF

The experiments described above show that the CUF can map challenges to responses that are easy to evaluate but hard to predict, which is in line with the definition of a physical random function[31]. Moreover, a CUF can neither be re-created, nor copied without physical access and detailed knowledge of the handle segments. Nevertheless, for improved cryptographic security, irreversibility and inherent non-copiability are desirable. Specifically, this means that (1) the input must not be accessible by reverse-processing the output information, (2) the brute force approach needs to be prohibitively resource-intensive, (3) the pool information must not be readable in its entirety, and (4) the CUF cannot be copied even by an actor with physical access and further information available. The first two points are fulfilled by the way CRPs are generated and processed, as well as the prohibitively large size of the orDNA pool, which are discussed in more detail in Supplementary Note 11.

Nonetheless, as it could be argued that technological advances might allow a future adversary to copy and sequence the entire CUF via the handle sequences to duplicate the pool and extract all possible CRPs, further measures were taken to prevent this. Notably, restriction sites were built into the outer handle sequences (Supplementary Fig. 2). The handles are still needed for CUF generation but can afterwards be cleaved. The restriction digest leaves only 7 and 6 defined bases and a 5' overhang at either side, with the outmost base being degenerate (Fig. 4a). Further, the 5'-overhangs are blunted using 2',3'-dideoxy degenerate bases. In combination, these chemical modifications prevent pool-wide PCR and ligation (Supplementary Note 12). The results of a library preparation protocol conducted by a third-party provider confirmed that the pool with the 2',3'-dideoxy ends can no longer be ligated (Fig. 4b).

In addition, the issuer of a CUF can, at least approximately, decide on the allowed number of executions by the number of copies made before implementing the modification, since replication and sequencing of the entire pool are no longer feasible by any straightforward means. Any attempt to do so would alter the chemical composition of the pool and thereby potentially harm the integrity of the function.

Even if a methodology were found by which sequencing can be achieved, reading and re-synthesizing $2.6 \cdot 10^{10}$ unique sequences would be prohibitively expensive. According to the coupon collector's problem, to read all sequences at least once, an average sequencing depth of $\ln(n) = 24$ is required, corresponding to about 625 billion reads, equivalent to reading ~1300 human genomes at equal depth. Assuming the lowest price range of 0.00001 $/bp[32], re-synthesis of 26 billion specific 100 nt sequences would additionally amount to a price of more than 25 million USD, compared to a reported black market price range of 4–30 USD for a hacked password[33]. This prohibitive time and cost argument is in line with the definition of unclonability as established in PUF literature, which refers to the requirement that it must be hard (in practice meaning expensive, as opposed to easy or low overhead cost) to construct another function that produces the same challenge response pairs[25].

As the cost and complexity increases with the pool size, a smaller, non-modified and therefore more readily invertible CUF could also work in analogy to moderately hard functions, which have been suggested to add a price tag to a given transaction[34].

## Discussion

The demonstrated scalability and distributability, as well as the unclonability, enable use cases similar to some of the applications of cryptographic hash functions or PUFs. DNA CUFs can be generated in near-unlimited quantities, as each random synthesis will create a new unique masterpool, from which a multitude of CUFs can be generated at low cost (a cost analysis is provided in Supplementary Note 13). Unlike mathematical functions or PUFs, CUFs (in their uncopiable state) only allow a certain amount of operations as decided in advance by the issuer, via the amount of copies distributed to each user. This is advantageous in terms of security, as it means that even with the correct CUF at hand and unlimited time to attempt a brute force attack, a potential adversary is highly unlikely to guess any CRP or password when the number of operations is intrinsically limited.

On the application side, CUFs are capable of performing similar tasks as suggested for PUFs, for example an authentication as described by McGrath et al.[35] (Fig. 4c). However, a PUF only consists of a single token, meaning only one simultaneous readout is possible. For authentication purposes, CRPs must therefore be recorded in advance. The thus generated database is used to validate the PUF's identity. In this case, the security of the system requires that the database is kept secret and that the authenticator is a trusted party. Such a typical use case could also be implemented using a CUF. In addition, the distributability of CUFs enables authentication within a group of equal users (Fig. 4d). This decentralized approach requires less trust in a single party and eliminates the risk associated with keeping a CRP database. CUFs can thus bridge the gap between physically unique PUFs and distributable mathematical algorithms.

Moreover, as recording of all possible CRPs is infeasible in terms of time and cost, and the number of supported CRPs grows exponentially with the number of random bases in the pool, CUFs currently fulfill the definition of a strong PUF[35]. Potential future security risks that can be identified are copying, sequencing and the brute force approach, as discussed above. However, with the vast pool complexity and the implemented modifications only significant technological advances, e.g., low-cost solid-state nanopore sequencing, would seriously reduce the cost and complexity of such attacks. Otherwise, much like with other technologies, the human factor is likely the weakest link[36], with social engineering attacks being successfully used to circumvent otherwise secure systems, including decentralized technologies such as cryptocurrencies[37].

While the relatively slow readout is an advantage in terms of security against brute force attacks, the time lag between entering the input and receiving the output is at the current state of the technology a clear drawback for using a CUF for repeated everyday use. There

may, however, be scenarios where the additional security stemming from its materiality and distributability is more critical than immediate readout. For example, real-life applications in anti-counterfeiting and the implementation of non-fungible items and product-integrated CUFs are conceivable in the near future.

In conclusion, we propose the concept of CUFs that use large operable DNA pools with random components to perform a computational task. CUFs are capable of returning an output from a given input, whereby the functionality is derived from the entropy of a chemical process, namely random DNA synthesis. The input-output conversion has an intrinsic cryptographic security comparable to PUFs by its irreversibility, the large size and diversity of the used DNA pools and the chemical modifications in place to prevent conceivable attacks. In alignment with physical unclonable functions, new functions can be generated on demand, which are impossible to duplicate from scratch by current means. Use cases that can tolerate a delay between input and readout can already be realized at the current state of implementation, and as the field of DNA information technology advances further, it is conceivable that hybrid systems using DNA in an interface with a classical computer could allow applications with real-time readout, combining speed with cryptographic security. DNA-based chemical unclonable functions thus have the potential to combine cryptography and chemical computation, not only on a conceptual level, but in practice as well.

## Methods

### Primers and datasets

A list of all primer sequences and library designs can be found in Supplementary Table S3. All DNA was ordered in dried state from Microsynth AG (Balgach, Switzerland). Furthermore, a list summarizing all experiments (39 main datasets + 15 extended datasets) with the used orDNA pools, input primers and cross-references to sequencing files can be found in Supplementary Table S4.

### Function synthesis/pool generation

A 5 nmol aliquot of the library as received from the supplier was dissolved in PCR-grade water (type 1, $18.2 \, M\Omega \, cm$ at 24 °C, Milli-Q®; Merck, Darmstadt, Germany). Dilution series were performed to achieve the concentrations needed to pipette the amount corresponding to the desired pool size ($10^8$–$2.6 \cdot 10^{10}$ individual sequences). Aside from the template, the final PCR mix contained 1x KAPA SYBR FAST qPCR master mix (KAPA Biosystems, Wilmington, USA) and $0.5 \, \mu M$ of each primer (Microsynth AG, Balgach, Switzerland). Depending on the pool size the final reaction volume comprised 20–80 µl. Dilutions and reaction mixes were prepared under laminar flow. Primers fw2 and rx (outer handles) were used as forward and reverse primers for amplification, respectively. Thermal cycling consisted of 180 s pre-incubation at 95 °C, followed by cycles of 15 s denaturing at 95 °C, 30 s annealing at 56 °C and 30 s elongation at 72 °C. The reaction was completed with 180 s of final elongation at 72 °C (cycling was stopped as soon as the fluorescence curves reached a plateau).

### Function propagation

Reaction mixes contained 10 µl KAPA SYBR FAST qPCR master mix (KAPA Biosystems, Wilmington, USA), 1 µl fw2 and rx primers (10 µM), 1 µl purified product of the previous "generation" and 7 µl PCR-grade water. Thermal cycling consisted of 180 s pre-incubation at 95 °C, followed by cycles of 15 s denaturing at 95 °C, 30 s annealing at 62 °C and 30 s elongation at 72 °C with 180 s final elongation. Cycling was stopped as soon as fluorescence reached a plateau.

### Selection PCR/function operation

A list of all PCR parameters and measured Ct-values can be found in Supplementary Table S5.

**Pool size $10^8$.** Reaction mixes with a total volume of 20 µl contained 10 µl KAPA SYBR FAST qPCR master mix (KAPA Biosystems, Wilmington, USA), 1 µl of the desired forward and reverse input primer (10 µM), respectively, 1 ng template (1 ng/µl) and 7 µl PCR-grade water. Thermal cycling consisted of 180 s pre-incubation at 95 °C, followed by cycles of 15 s denaturing at 95 °C, 30 s annealing at 62 °C and 30 s elongation at 72 °C with 180 s final elongation. For input primers with a GC-content below 40%, the annealing temperature was reduced to 56 °C.

**Pool size $1.6·10^9$.** Reaction mixes with a total volume of 20 µl contained 10 µl KAPA SYBR FAST qPCR master mix (KAPA Biosystems, Wilmington, USA), 1 µl of forward and reverse input primer (10 µM), respectively, 0.4 ng template and 7 µl PCR-grade water. Thermal cycling consisted of 180 s pre-incubation at 95 °C, followed by cycles of 15 s denaturing at 95 °C, 30 s annealing at 62 °C and 30 s elongation at 72 °C with 180 s final elongation. Cycling was stopped as soon as fluorescence reached a plateau.

**Pool size $2.6·10^{10}$.** Reaction mixes with a total volume of 50 µl contained 25 µl KAPA SYBR FAST qPCR master mix (KAPA Biosystems, Wilmington, USA), 2.5 µl of forward and reverse input primer (10 µM), respectively, 23 ng template and 18 µl PCR-grade water. Thermal cycling consisted of 180 s pre-incubation at 95 °C, followed by cycles of 15 s denaturing at 95 °C, 30 s annealing at 62 °C and 30 s elongation at 72 °C with 180 s final elongation. Cycling was stopped as soon as fluorescence reached a plateau.

### Trimming PCR
Reaction mixes contained 10 µl KAPA SYBR FAST qPCR master mix (KAPA Biosystems, Wilmington, USA), 1 µl of forward and reverse primers (0F, rv2, both 10 µM), 1 ng template (DNA purified from selection PCR) and 7 µl PCR-grade water. Thermal cycling consisted of 180 s pre-incubation at 95 °C, followed by cycles of 15 s denaturing at 95 °C, 30 s annealing at 56 °C and 30 s elongation at 72 °C with 180 s final elongation. Cycling was stopped as soon as fluorescence reached a plateau.

### Illumina sequencing
Following trimming, sequencing adapters were added by two subsequent PCR reactions. Both reactions were run at equal conditions, with reaction mixes containing 10 µl KAPA SYBR FAST qPCR master mix (KAPA Biosystems, Wilmington, USA), 1 µl of forward and reverse primers (10 µM), 1 ng template (purified DNA from previous step) and 7 µl PCR-grade water. Thermal cycling consisted of 180 s pre-incubation at 95 °C, followed by cycles of 15 s denaturing at 95 °C, 30 s annealing at 56 °C and 30 s elongation at 72 °C with 180 s final elongation. Cycling was stopped as soon as fluorescence reached a plateau. The first reaction used primers 1R-AL and 1F, the second reaction the Illumina primers 2FU and an indexed reverse primer (2RI). Both reactions were gel-purified (refer to "work-up of PCR reactions"). Quality control gels of the samples after sequencing preparation are shown in Supplementary Figs. 13–17. The samples were sequenced on an iSeq 100 system (Illumina, San Diego, CA, USA) at a final concentration of 50 pM containing 2% PhiX reference.

### Work-up of PCR reactions
Following preparative PCR reactions, DNA purification and work-up was performed using the DNA Clean & Concentrator kit (Zymo Research, Irvine, CA, USA) according to the manufacturer's protocol. The purified product was eluted in PCR-grade water (type 1, 18.2 MΩ cm at 24 °C, Milli-Q®; Merck, Darmstadt, Germany). Analytical agarose gel electrophoresis was performed to confirm product size, using E-Gel EX gels (2% or 4%, Invitrogen, Thermo Fisher Scientific) on a Power Snap Electrophoresis Device (Thermo Fisher

Scientific). The same system and conditions were used for preparative AGE purification during sequencing preparation, whereby the desired bands were excised and the DNA extracted using a Zymoclean Gel DNA Recovery Kit (Zymo Research, Irvine, CA, USA) according to the manufacturer's protocol. DNA concentrations were measured using Qubit fluorometric quantification (Thermo Fisher Scientific, Waltham, MA, USA).

### Restriction digest
For restriction digestion to remove the outer handles, the 50 µl reaction mix consisted of 1 µg DNA, 1x rCutSmart buffer, 50 U PleI enzyme (concentrated at 5U/µl) in PCR-grade water. The buffer and enzyme were obtained from New England Biolabs (Ipswich, MA, USA). All components were mixed on ice with the enzyme added last, then incubated at 37 °C for 70 min. Analytical agarose gel electrophoresis was performed using E-Gel EX gels (2%, Invitrogen, Thermo Fisher Scientific) on a Power Snap Electrophoresis Device (Thermo Fisher Scientific).

### End inertization
The product digested with PleI (producing a 5' single nucleotide overhang of each base) was treated with Sequenase (Thermo Fisher Scientific, Waltham, MA, USA) and degenerate 2'3'-dideoxy nucleotides (ddNTPs, New England Biolabs Ipswich, MA, USA) for 3' recessed end fill-in. The reaction containing 300 ng of digested DNA was performed in 1X reaction buffer provided with the polymerase, at a 300 µM degenerate ddNTP concentration and with of 13U of the enzyme, in a total volume of 100 µl. The mix was incubated for 1 min at 37 °C, and purified using the DNA Clean & Concentrator kit (Zymo Research, Irvine, CA, USA) according to the manufacturer's protocol. As a control, the analogous reaction using dNTPs instead of ddNTPs was performed.

### Ligation test
In total, 200 ng of a PleI-digested and ddNTP-blunted orDNA library were sent to an external service provider (FASTERIS SA, Plans-les-Ouates, Switzerland) to perform Illumina library preparation with two-sided ligation of adapter sequences. As positive controls, an untreated orDNA pool and an orDNA-pool digested with PleI and blunted with dNTPs instead of ddNTPs were provided along with the test sample. Results obtained from FASTERIS SA comprised quality control data in the form of electropherograms and gel images of the test sample and the control pre- and post-ligation.

### Data processing
Python scripts were used for filtering, read frequency analysis, as well as visualization of the positional base content of the FASTQ files obtained from Illumina sequencing. Overall position-dependent base counts were plotted using a MATLAB script. A pipeline consisting of read filtering and frequency analysis using BBMap (v38.99), as well as custom Python scripts for k-mer extraction, weighted Jaccard similarity calculation, MinHash signature generation, fuzzy extraction, and parameter optimization was used.

### Reporting summary
Further information on research design is available in the Nature Portfolio Reporting Summary linked to this article.

## Data availability
All raw sequencing data generated in this study have been made public on the European Nucleotide Archive under Accession Code PRJEB73810. The literature data from Meiser et al.[11] used in this study are available in the Figshare database under https://figshare.com/articles/dataset/SequencingData/12941786/1. Source data are provided with this paper.

## Code availability

Annotated code used for data processing and analysis is available on Github (github.com/fml-ethz/cuf-cryptography).

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

## Acknowledgements

This project was financed by the European Union's Horizon 2020 Program, FET-Open: DNA-FAIRYLIGHTS, Grant Agreement No. 964995 and ETH Zurich.

## Author contributions

A.M.L.: conceptualization, methodology, investigation, formal analysis, data curation, writing—original draft, software, visualization, A.L.G.: software, formal analysis, data curation, writing—review and editing, W.J.S.: conceptualization, resources, R.H.: methodology, software, formal analysis, data curation, writing—review and editing, R.N.G.: conceptualization, methodology, resources, supervision, software, writing—review and editing, funding acquisition.

## Funding

## Competing interests

A.M.L., W.J.S., R.H. and R.N.G. declare a financial interest as inventors of a patent application (PCT/EP2022/059323), filed by ETH Zurich and TU Munich, covering the invention of operable random DNA. A.L.G. does not declare any conflict of interest.
