## [Peer Review File · Nature Communications]

Reviewers' Comments:

Reviewer #1:

Remarks to the Author:

This is a well written paper on a very interesting and exciting topic. It is based on a simple yet smart idea and demonstrates the technology using an in vitro case study followed by thorough analyses of the data. The paper also discusses in detail broader implications and potential use cases. I am in favor of publishing assuming the following comments are addressed:

1. PUFs are (by definition) robust, unique and unclonable. I see that CUFs are robust. How about uniqueness? I did not see proof that a CUF cannot be re-created. How do you establish unclonability?
2. Figure 3a. It was not clear how the challenge sequences/primers were selected. How do these sequences rank (eg freq) in the overall CUF pool? Does it make a difference in the challenge/response performance? At least 3 should be performed.
3. Figure 3b. I expect these sequential pcr amplifications to impact the distributions in the CUFs. Did you sequence these P1-5 pools?
4. Figure 3b. How did you pick the challenge? Is the performance equally good for other challenges? At least 3 should be performed.
5. The MinHash/Fuzzy extractor approach is somewhat of an overkill for the examined data set. A simple filter would do the job here, as the noise level is low. It could potentially be useful when applying the challenge to P1 all the way to P5. Can you demonstrate that the classification is correct (and binary) in all cases (versus the similarity score which clearly drops for P4 and P5)?
6. Figure 4a/b. I see this a motivating example, but can you really claim that the CUF pool cannot be sequenced? What is preventing me from cutting (randomly) and attaching probes that will then allow me to sequence? This would not be that difficult to do.
7. The paper cites very few references, missing substantial results from DNA computing and PUFs. At minimum the following should papers should be be cited:

DNA computing:

<https://www.nature.com/articles/35106533>

<https://www.nature.com/articles/s41586-023-06484-9>

PUFs: <https://eprint.iacr.org/2009/277>

Genetic PUFs: <https://www.science.org/doi/10.1126/sciadv.abm4106>

Reviewer #2:

Remarks to the Author:

The manuscript describes the use of partially randomized libraries of double-stranded DNA molecules as a means of generating "chemically unclonable functions" mapping inputs to outputs via PCR reactions, including the application of standard hashing algorithms for cryptographic key generation. This has potential applications in the development of "unclonable" physical objects that could be used as secure keys or key generators.

In general, the manuscript is well written and the results presented demonstrate the utility of this PCR-based approach, though this is perhaps not a surprise given the robustness and well studied nature of PCR as an experimental technique.

The question of irreversibility and unclonability is perhaps the weakest aspect of the work. The authors claim that cleaving the the ends of the handles to leave an overhang and modifying the 5' overhangs to produce dideoxy bases suffices to prevent their ligation into a blunt-end vector for subsequent cloning. This discussion neglects the possibility of nanopore sequencing being used to sequence the molecules from a sample of the solution. The use of a commercial synthesis provider to generate the random sequences has potential implications for the overall security of this technology, as the provider could simply retain a sample of the solution, but this appears to be mitigated by the selection of a subset of the "master pool" for use as a CUF. However, this does not seem to be discussed as a motivation for downselecting the CUF from the master pool in the manuscript text.

Given the claims that are made of this system as the potential basis of physically secure

cryptographic systems, a more detailed discussion of the security implications of the system might be expected.

Some minor comments on the presentation:

Fig 1b - it would be preferable to clarify that this is not a novel research contribution but rather just an illustration of standard commercial solid-state synthesis approaches for partially randomized sequences.

Fig 2 - the accompanying text refers to "qualitative visual inspection" of the results obtained from various challenges; it might be good to pre-empt reader questions about the qualitative nature of this data by referring forward to Figure 3, which contains the necessary quantitative analysis.

Reply to the referees' comments

Reviewer 1

We appreciate the careful assessment and would like to thank reviewer 1 for the work on our manuscript, which we revised accordingly. We think that the changes and additional data requested by the reviewer have improved the overall quality of the manuscript.

Comments

1. PUFs are (by definition) robust, unique and unclonable. I see that CUFs are robust. How about uniqueness? I did not see proof that a CUF cannot be re-created. How do you establish unclonability?

We appreciate this comment and see that the manuscript did not sufficiently address this, as it was a point raised by both reviewers.

Uniqueness, as it is defined in e.g. Li et al.¹, refers to the fact that an unclonable function's mapping does not coincide with the mapping of any other identically manufactured function. This is given by the random manufacturing process and was experimentally confirmed by running the same challenges with different DNA pools (Fig. 3a, experiments 1-4, 20-25), showing that this does not lead to collisions.

As for unclonability, this is defined such that it must be *hard* (as opposed to *easy*, which is in practice defined as involving low overhead cost) to construct another PUF Π_r such that $\forall x \in X : \Pi_r(x) \approx \Pi(x)$ up to a small error.² There are two potential ways in which someone may attempt to clone a CUF, and as described below and in the manuscript text, these are practically infeasible and suffice the above definition of *hard*. The first is reading all the sequences in the pool and then de-novo synthesizing. This is first and foremost prevented by the sheer entropy and size of the pool (sequencing and re-synthesizing a pool of >10 billion sequences would cost millions of dollars, and scales with increased size and entropy). While random DNA pools can be synthesized at low cost, this is not at all true for the de-novo synthesis of sequence defined DNA pools, which would be required in a cloning attempt and are not accessible at this scale. This argument of prohibitive time and cost constraints is commonly used to assert a PUF's unclonability, as it can rarely be disproven without a doubt that, given enough resources combined with technical advances, a token can be sufficiently analyzed and reproduced. Notably, the same argument is used for CRISPR-PUFs¹. In addition, in our implementation we introduce further obstacles by the 3'-end modification. This renders sequencing even more difficult, as common protocols to introduce the necessary adapters can no longer be implemented.

The absence of usable primer regions after removal of the end sequences also impedes the second possibility to clone the function, i.e. creating identical copies the pool by PCR as by definition PCR requires constant end regions to allow for amplification.

To add this valuable discussion and improve clarity, we made amendments to the manuscript to reflect the abovementioned points.

Specifically, we now more explicitly state the time and cost constraints:

"[...]synthesizing the same number of determined sequences would be infeasible. Companies specializing in synthesis of large DNA pools currently offer prices of approx. 50'000 USD for a batch of a million sequence defined oligos, of which several billion of batches would be required to amount to $6 \cdot 10^{16}$ sequences. This asymmetry in time and cost means that no randomly synthesized masterpool can be re-created with intent.."

This is then further elaborated in two later sections:

"Consequently, the combination between the pool's intrinsic randomness and the subsequent chemical edits guarantees the unclonability of the function. Re-creating the same pool from scratch would require knowledge of the entire composition as generated by the random manufacturing process, followed by controlled re-synthesis of all unique sequences. Both are prohibitively time-

consuming and costly due to the large pool size, and additionally rendered difficult to impossible by the truncated ends that prohibit global PCR and ligation.”

“[The] prohibitive time and cost argument is in line with the definition of unclonability as established in PUF literature, which refers to the requirement that it must be hard (i.e. expensive, as opposed to “easy” or low overhead cost) to construct another function that produces the same challenge response pairs.²”

In addition, these aspects are discussed in further detail in Supplementary Note 12.

We also added a section to the aforementioned note:

“The modification additionally introduces obstacles to simultaneous sequencing of the entire pool, as this typically involves library preparation by adding adapters, either by PCR or by ligation. This also includes “PCR-free” methods, such as nanopore sequencing. The truncated and modified 3'-ends would need to be circumvented, which, even if possible, would increase the complexity and cost of sequencing and copying further.”

“In conclusion, it can be shown that, aside from the necessity of gaining access to the pool with malicious intent, there is no straightforward way to read and copy the pool. Any such attempt would increase the effort, cost and complexity, and introduce the risk of altering the pool in a way that hampers its performance as a random function.”

2: Figure 3a. It was not clear how the challenge sequences/primers were selected. How do these sequences rank (eg freq) in the overall CUF pool? Does it make a difference in the challenge/response performance? At least 3 should be performed.

We assessed >50 challenge-response-pairs, whereby several challenges were performed multiple times ($n \geq 3$), in addition to the replicates using the P1-P5 generations as templates. Among the many challenges posed to the systems not a single duplicate led to a false positive or false negative result.

The first experiments were conducted with a randomly selected challenge sequence. The following challenges comprised small permutations of the same challenge to test the system’s robustness (e.g. experiments 5-8, 13-16 in Figure 3a). As shown in the data, even small permutations (single base change in the challenge) created a completely new output.

Regarding the frequency in the pool, this can only be answered in terms of statistics, as, by design, the pool composition can never be fully known as it has been created by random synthesis (a true random process). This means that the composition of the pool is also not known by the creator of the pool. Expected frequencies are discussed in Supplementary Notes 3 and 4 (with complementary Supplementary Fig. 7) along with more detailed considerations regarding primer design in relation to the pool size (please also refer to reply to comment 3).

To additionally clarify this in the main manuscript, we also added a sentence:

“While the input sequence was chosen at random, the number of input bases within this initial primer pair was chosen such that given the pool size, the expectation value of the frequency of perfectly matching sequences in the pool was approx. 1.5 (see Supplementary Note 4).”

3. Figure 3b. I expect these sequential pcr amplifications to impact the distributions in the CUFs. Did you sequence these P1-5 pools?

P1-5 pools were not sequenced. As pointed out in the manuscript and is a key property of the pools used in this work, the pools are too large to be sequenced. Due to the random synthesis process, the pools P1-P5 comprise 100 million unique sequences each. Although they are two orders of magnitude smaller than some other pools that we tested in this work, sequencing them at the depth required to read all sequences and be able to quantify bias specifically caused by PCR, is infeasible at the lab scale. Therefore, the only way for us to evaluate if the P1-5 pools were affected by PCR amplification is by testing if the same inputs yield the same outputs, which is true as shown in Figure 3c.

In addition, we also conducted more experiments with P1-P5, which are shown in Supplementary Figure 4 (see also reply to comment 4).

4. Figure 3b. How did you pick the challenge? Is the performance equally good for other challenges? At least 3 should be performed.

The same randomly picked challenge C1 was used as for the initial experiments. This rationale is now stated more explicitly in the manuscript (see below).

As requested by the referee, and to further show the robustness of the pool during PCR, we also generated additional data of proliferation results for three more challenges in Supplementary Fig. 4.

Among them were also two challenges that only differ by a single base ($D_L = 1$) relative to C1. These additional data were generated 20 months after the first readout. Even under these conditions, the distributions are still clearly separated, as shown in Supplementary Fig. 4b-c

In accordance, we re-worked the manuscript to now specify:

“Figure 3c shows the consistency of the output throughout the daughter generations, whereby the same randomly chosen base challenge (C1) was used as an input as for the initial replicates (further challenge-response pairs are shown in Supplementary Figure 4). The resulting outputs still showed high similarities across all generations, as opposed to e.g. comparing outputs resulting from inputs differing by a single base. While any PCR bias introduced throughout the proliferation steps cannot be quantified due to the pools being too large to be sequenced within reasonable time and cost constraints, the robustness in terms of differentiating input-output similarity is maintained across the generations.”

5. The MinHash/Fuzzy extractor approach is somewhat of an overkill for the examined data set. A simple filter would do the job here, as the noise level is low. It could potentially be useful when applying the challenge to P1 all the way to P5. Can you demonstrate that the classification is correct (and binary) in all cases (versus the similarity score which clearly drops for P4 and P5)?

We agree with the referee that the MinHash/Fuzzy extractor approach is quite elaborate. However, an elaborate method is required to enable the evaluation of correct hash values without requiring a comparison of the outputs on the basis of a similarity score. This step allows to make a binary decision (true/false) of whether a key is correct or false. Since the sequencer output is a set of noisy sequences, we don't see how this decision can be made with a simple filter, even though the noise level is rather low. The results of our approach are shown in Figures 3e, 3f and 3g, showing that all outputs were correctly made and are binary as belonging to like or unlike inputs, which includes the ones from P1 to P5 shown in Figure 3c. Note that our hashing approach is rather simple and computationally very cheap, so we don't see an issue with this method.

6. Figure 4a/b. I see this a motivating example, but can you really claim that the CUF pool cannot be sequenced? What is preventing me from cutting (randomly) and attaching probes that will then allow me to sequence? This would not be that difficult to do.

As stated above (reply to comment 1), sequencing of the pool in its entirety is mainly prevented by the time and cost constraints (as is the brute force approach), as hundreds of billions of reads would be required. The additional end modification further adds to this difficulty, and preventing adapter ligation also precludes reliable copying of the entire pool by PCR once the modification has been implemented.

The cost constraint of the massive number of sequences that need to be read also holds in case of successful random fragmentation and probe attachment. Moreover, fragmentation would increase the required read count further, and add additional costs for the probes.

We now added the following passage to the main manuscript, going in a bit more detail regarding cost estimates and also including the concept of a pricing function to our discussion:

“In addition, the issuer of a CUF can, at least approximately, decide on the allowed number of executions by the number of copies made before implementing the modification, since replication and sequencing of the entire pool are no longer feasible by any straightforward means. Any attempt to do so would alter the chemical composition of the pool and thereby potentially harm the integrity of the function. Even if a methodology were found by which sequencing can be achieved, reading and re-

synthesizing $2.6 \cdot 10^{10}$ unique sequences would be prohibitively expensive. According to the coupon collector's problem, to read all sequences at least once, an average sequencing depth of $\ln(n) = 24$ is required, corresponding to about 625 billion reads, equivalent to reading approx. 1300 human genomes at equal depth. Assuming the lowest price range of 0.00001 \$/bp³, re-synthesis of $2.6 \cdot 10^{10}$ sequence specific 100 nt sequences would additionally amount to a price of more than 25 million USD, compared to a reported black market price range of 4-30 USD for a hacked password.⁴ This prohibitive time and cost argument is in line with the definition of unclonability as established in the PUF literature, which refers to the requirement that it must be hard (i.e. expensive, as opposed to "easy" or low overhead cost) to construct another function that produces the same challenge response pairs.²

As the cost and complexity increases with the the number of random bases in the pool, a smaller, non-modified and therefore more readily invertible CUF could also work in analogy to moderately hard functions, which have been suggested to add a price tag to a given transaction⁵."

7. The paper cites very few references, missing substantial results from DNA computing and PUFs. At minimum the following should papers should be be cited:

DNA computing:

<https://www.nature.com/articles/35106533>

<https://www.nature.com/articles/s41586-023-06484-9>

PUFs: <https://eprint.iacr.org/2009/277>

Genetic PUFs: <https://www.science.org/doi/10.1126/sciadv.abm4106>

We thank the reviewer for the additional literature suggestions and cited the suggested references 1, 2 and 4 (3 was already cited in the previous version).

We added the following statements and extended information to the introduction to accommodate and comment on the newly introduced references:

"Even DNA-based programmable gate arrays for general purpose computing have been introduced⁶, as well as programmable automata⁷."

"Genetic information has already been suggested as a medium for physical unclonable functions by using CRISPR-induced nonhomologous end joining repair to generate a unique barcode-indel mapping (CRISPR-PUFs).¹ There, the random process refers to the combination of barcodes and indels in a given cell line. We instead propose to directly use of randomly generated DNA sequence, giving rise to massive levels of entropy."

Reviewer 2

We appreciate reviewer 2's detailed questions and inputs and addressed them. The comments were a valuable contribution to improving the quality of the manuscript.

Comments

1: The question of irreversibility and unclonability is perhaps the weakest aspect of the work. The authors claim that cleaving the ends of the handles to leave an overhang and modifying the 5' overhangs to produce dideoxy bases suffices to prevent their ligation into a blunt-end vector for subsequent cloning. This discussion neglects the possibility of nanopore sequencing being used to sequence the molecules from a sample of the solution.

As similar concerns regarding unclonability were raised by reviewer 1, we took this very seriously and took efforts to address this.

We now make it more clear that the main protection against irreversibility and unclonability is the massive number of sequences, which puts a very high price tag in terms of financial and labor efforts to any attempt in doing so. Other PUFs are considered unclonable for the same reasons¹. The end modification is an additional factor that significantly increases the effort.

We re-worked the manuscript by adding the following:

"In addition, the issuer of a CUF can, at least approximately, decide on the allowed number of executions by the number of copies made before implementing the modification, since replication and sequencing of the entire pool are no longer feasible by any straightforward means. Any attempt to do so would alter the chemical composition of the pool and thereby potentially harm the integrity of the function. Even if a methodology were found by which sequencing can be achieved, reading and re-synthesizing $2.6 \cdot 10^{10}$ unique sequences would be prohibitively expensive. According to the coupon collector's problem, to read all sequences at least once, an average sequencing depth of $\ln(n) = 24$ is required, corresponding to about 625 billion reads, equivalent to reading approx. 1300 human genomes at equal depth. Assuming the lowest price range of 0.00001 \$/bp³, re-synthesis of 26 billion specific 100 nt sequences would additionally amount to a price of more than 25 million USD, compared to a reported black market price range of 4-30 USD for a hacked password.⁴ This prohibitive time and cost argument is in line with the definition of unclonability as established in PUF literature, which refers to the requirement that it must be hard (i.e. expensive, as opposed to "easy" or low overhead cost) to construct another function that produces the same challenge response pairs.²

As the cost and complexity increases with the the number of random bases in the pool size, a smaller, non-modified and therefore more readily invertible CUF could also work in analogy to moderately hard functions, which have been suggested to add a price tag to a given transaction⁵."

In a later section we again come back to this:

"Consequently, the combination between the pool's intrinsic randomness and the subsequent chemical edits guarantees the unclonability of the function. Re-creating the same pool from scratch would require knowledge of the entire composition as generated by the random manufacturing process, followed by controlled re-synthesis of all unique sequences. Both are prohibitively time-consuming and costly due to the large pool size, and additionally rendered difficult to impossible by the truncated ends that prohibit global PCR and ligation."

Regarding nanopore sequencing, the cost would be comparable, if not higher⁸, due to the short read length (as opposed to long sequences, for which nanopore is better suited). Moreover, nanopore sequencing typically requires the addition of adapters by ligation, which is prevented by the truncated 3'-ends. Any attempt to introduce adapters by different means would complicate the process and again potentially alter the pool. Furthermore, very high concentrations and copy numbers are required for successful reading through the pores, both of which are not given since effortless copying by PCR is no longer feasible.

Prompted by the reviewer's mention of nanopore sequencing we therefore added an additional section regarding this to supplementary note 12:

"The modification additionally introduces obstacles to simultaneous sequencing of the entire pool, as this typically involves library preparation by adding adapters, either by PCR or by ligation. This also

includes “PCR-free” methods, such as nanopore sequencing. The truncated and modified 3'-ends would need to be circumvented, which, even if possible, would increase the complexity and cost of sequencing and copying further.”

“In conclusion, it can be shown that, aside from the necessity of gaining access to the pool with malicious intent, there is no straightforward way to read and copy the pool. Any such attempt would increase the effort, cost and complexity, and introduce the risk of altering the pool in a way that hampers its performance as a random function.”

2: The use of a commercial synthesis provider to generate the random sequences has potential implications for the overall security of this technology, as the provider could simply retain a sample of the solution, but this appears to be mitigated by the selection of a subset of the “master pool” for use as a CUF. However, this does not seem to be discussed as a motivation for downselecting the CUF from the master pool in the manuscript text.

As the reviewer correctly noted, an external supplier is a potential security risk, which we had not considered in the manuscript. We now stated clearly that sub-sampling indeed mitigates that risk, by adding the following sentence to the main manuscript:

“This sub-selection step represents an additional random process (urn sampling without replacement), ensuring that not even an actor having the masterpool can re-create a given subpool.”

In addition, with the availability of lab-scale synthesis systems, the necessity of an external provider can be eliminated altogether.

3: Given the claims that are made of this system as the potential basis of physically secure cryptographic systems, a more detailed discussion of the security implications of the system might be expected.

We agree that this discussion turned out a bit short. While our study did not go into depth to assert cryptographic security, we can compare the presented CUFs to existing PUF literature, showing that in principle, the CUFs fulfil the criteria established for so-called strong PUFs (as opposed to weak PUFs, which still provide a level of security, but have a more narrow scope of application)⁹. Furthermore, we can discuss the future technological requirements that would have to be met for a successful attack. We now re-worked and added to a section in the discussion to accommodate this:

“Moreover, as recording of all possible CRPs is infeasible in terms of time and cost, and the number of supported CRPs grows exponentially with the the number of random bases in the pool, CUFs currently fulfil the definition of a strong PUF. Potential future security risks that can be identified are copying, sequencing and the brute force approach, as discussed above. However, with the vast pool complexity and the implemented modifications only significant technological advances, e.g. low-cost solid-state nanopore sequencing, would seriously reduce the cost and complexity of such attacks. Otherwise, similar to other technologies, the human factor is likely the weakest link¹⁰, with social engineering attacks being successfully used to circumvent otherwise secure systems, including decentralized technologies such as cryptocurrencies¹¹.”

Further, we made a minor adjustment to set the security in context with comparable PUFs:

“The input-output conversion has an intrinsic cryptographic security comparable to PUFs by its irreversibility, the large size and diversity of the used DNA pools and the chemical modifications [...]”

In addition, we discuss brute force attacks and reverse-engineering of CRPs in depth in supplementary note 11.

Moreover, security is also linked to the aspect of unclonability (in case an adversary gets access to the token for an extended period of time). As pointed out in our reply to comment 1, this discussion is now also expanded. The same goes for the security implications for choosing an external provider, which we have now also addressed in the main text (see reply to comment 2).

Combined, we hope these changes add to the clarity of the manuscript and the strength, but also the differentiation, of our claims.

4: Fig 1b - it would be preferable to clarify that this is not a novel research contribution but rather just an illustration of standard commercial solid-state synthesis approaches for partially randomized sequences.

We thank the reviewer for the input and changed the corresponding figure caption to clarify this:

"Step-by-step chemical synthesis of orDNA, combining random with sequence-determined synthesis, as offered by many commercial suppliers."

5: Fig 2 - the accompanying text refers to "qualitative visual inspection" of the results obtained from various challenges; it might be good to pre-empt reader questions about the qualitative nature of this data by referring forward to Figure 3, which contains the necessary quantitative analysis.

We thank the reviewer for raising this clarity issue and as a corrective action re-wrote the corresponding section to make this more explicit. It now states:

"Even before quantitative analysis, comparison of two different challenges (each conducted twice) on the raw data level (after filtering for presence of expected constant regions), visualizing base content by position shows that two responses belonging to the same input/challenge are qualitatively similar, and responses to a different challenge lead to a different set of amplicons."

References in the rebuttal:

1. Li Y., Bidmeshki M. M., Kang T., Nowak C. M., Makris Y. & Bleris L. Genetic physical unclonable functions in human cells. *Sci. Adv.* **8**, eabm4106 (2022).
2. Maes R. & Verbauwhede I. A discussion on the Properties of Physically Unclonable Functions. In: *TRUST 2010 Workshop, Berlin* (2010).
3. Kosuri S. & Church G. M. Large-scale de novo DNA synthesis: technologies and applications. *Nat. Methods* **11**, 499-507 (2014).
4. Blocki J., Harsha B. & Zhou S. On the Economics of Offline Password Cracking. In: *2018 IEEE Symposium on Security and Privacy (SP)* (2018).
5. Dwork C. & Naor M. Pricing via Processing or Combatting Junk Mail. In: *Advances in Cryptology — CRYPTO' 92* (ed Brickell EF). Springer Berlin Heidelberg (1993).
6. Lv H., *et al.* DNA-based programmable gate arrays for general-purpose DNA computing. *Nature* **622**, 292-300 (2023).
7. Benenson Y., Paz-Elizur T., Adar R., Keinan E., Livneh Z. & Shapiro E. Programmable and autonomous computing machine made of biomolecules. *Nature* **414**, 430-434 (2001).
8. Bridge Informatics. Battle of the Sequencers: Illumina vs Nanopore Sequencing <https://bridgeinformatics.com/battle-of-the-sequencers-illumina-vs-nanopore-sequencing/> last accessed 07FEB2024.).
9. McGrath T., Bagci I. E., Wang Z. M., Roedig U. & Young R. J. A PUF Taxonomy. *Appl. Phys. Rev.* **6**, 11303 (2019).
10. Mitnick K. D. & Simon W. L. *The art of deception: Controlling the human element of security*. John Wiley & Sons (2003).
11. Weber K., Schütz A. E., Fertig T. & Müller N. H. Exploiting the Human Factor: Social Engineering Attacks on Cryptocurrency Users. In: *Learning and Collaboration Technologies. Human and Technology Ecosystems* (eds Zaphiris P, Ioannou A). Springer International Publishing (2020).

Reviewers' Comments:

Reviewer #1:

Remarks to the Author:

Thank you for responding to my comments.

Reviewer #2:

Remarks to the Author:

The authors have made a number of modifications to the manuscript which largely address the concerns raised by the original reviewers.

On the question of unclonability, the authors make an argument based largely on the high cost of sequencing and de-novo synthesizing every single molecule in the solution to create a precise replica. While that seems plausible given current technology, it is worth noting that sequencing technology has advanced substantially in recent years and I am not 100% convincing that the first reviewer's comment about being able to attach random primers and sequence has been quite addressed. (I don't think this is necessarily a barrier to accepting the manuscript, though.)

I am also slightly concerned that this response is rather focused on such "deterministic" attacks, for want of a better term for it. Given that the output from the CUF process is noisy data, how much noise can be tolerated? Does an attacker need to reproduce the precise molecules in the CUF to get below that margin, or is some error permissible? Given that random DNA pools can be synthesized easily and cheaply, as mentioned by the authors, is it possible that an attacker could do enough sequencing to figure out the probability distributions over bases used to create the initial random pool and then synthesize their own pool with similar probabilities? This could be done more cheaply than the authors imply here and with a large enough pool they could reproduce some of the same strands that were used in the CUF. With a plausible number of repeated attempts, could an attacker produce something that is "close enough" to a given CUF to fit below the error threshold?

Reviewer #1 (Remarks to the Author):

Thank you for responding to my comments.

Reviewer #2 (Remarks to the Author):

The authors have made a number of modifications to the manuscript which largely address the concerns raised by the original reviewers.

On the question of unclonability, the authors make an argument based largely on the high cost of sequencing and de-novo synthesizing every single molecule in the solution to create a precise replica. While that seems plausible given current technology, it is worth noting that sequencing technology has advanced substantially in recent years and I am not 100% convincing that the first reviewer's comment about being able to attach random primers and sequence has been quite addressed. (I don't think this is necessarily a barrier to accepting the manuscript, though.)

We agree with the reviewer that we cannot claim that the technology is future-proof, which CUFs have in common with other physical unclonable functions. We would also like to note that in general security depends on the state of current technologies. For example, many cryptography schemes rely on the difficulty of factorizing polynomials, however potential future quantum computers could render those schemes vulnerable.

Nonetheless addressing the question of sequencing, we observe that in the last ten years, the price reduction rate has slowed down and the sequencing price stabilized¹. Therefore, a major disruptive technological breakthrough would be necessary to make sequencing of the entire pool viable in terms of cost, not even considering the additional difficulty stemming from the truncated ends. Moreover, sequencing would be a necessary, but not a sufficient condition to reproduce a CUF. As we point out in the manuscript, aside from sequencing, synthesis cost is a major contributor to the overall price of recreating the CUF. To date, the largest synthetic DNA pools ever created are in the order of 10 million defined oligos (at significant costs), which is several orders of magnitude away from the 10 billion defined oligos required to recreate the CUFs described in the manuscript.

And finally, for common PUF applications where the token is a private key for the users, all these attacks are irrelevant, since physical access to the entire pool is a primary prerequisite to derive input-output pairs or any other information about it.

I am also slightly concerned that this response is rather focused on such "deterministic" attacks, for want of a better term for it. Given that the output from the CUF process is noisy data, how much noise can be tolerated? Does an attacker need to reproduce the precise molecules in the CUF to get below that margin, or is some error permissible? Given that random DNA pools can be synthesized easily and cheaply, as mentioned by the authors, is it possible that an attacker could do enough sequencing to figure out the probability distributions over bases used to create the initial random pool and then synthesize their own pool with similar probabilities? This could be done more cheaply than the authors imply here and with a large enough pool they could reproduce some of the same strands that were used in the CUF. With a plausible number of repeated attempts, could an attacker produce something that is "close enough" to a given CUF to fit below the error threshold?

The attacker does not need to reproduce the precise molecules in the CUF to 'break' the CUF, a small error is permissible, for example the error from PCR and sequencing is evidently permissible. We

have an indirect indication based on our data of how small the tolerated noise level is, i.e. smaller than one base difference in the input. This minimal Levenshtein distance on the input side already produces clearly distinct, but overall reproducible outputs.

This small tolerated error therefore doesn't challenge the security of the CUF since in order to reproduce the pool of DNA even approximately (i.e., allowing for some level of base errors), the attacker still has to sequence the entire pool or at the very least a large fraction of it at high accuracy. As the pools are large enough for the probability distributions of bases to be random, no useful information about the pool can be inferred from this. This is further strengthened by the fact that only a very small fraction of the possible sequence space is represented in the pool (currently up to ca 10^{10} sequences out of $4^{40} \approx 10^{24}$ possibilities in the proposed design). Thus, and as is stated in the manuscript, the CUF does not merely consist of a randomly synthesized pool, but also the random sub-selection from a much larger space of possibilities. In consequence, a draw comprising 10^{10} sequences from an equivalent but separately produced random synthesis (comprising the same base probabilities) would only contain very few of the same strands, if any at all.

And neither is it sufficient to just by luck reproduce a single (or a few) challenge-response pairs. For it to be a useful attack, the CUF needs to map a reasonable amount of all challenge-response-pairs. Our analysis shows that even for the smallest tested pool size, at least 26 bit (13 nts) inputs are possible, which translates to ca. 70 million challenge-response pairs for every CUF. Combining these factors, reproducing a CUF "by chance" (as opposed to reproducing a CUF through re-building it) is extremely unlikely and practically impossible. To make this point more clear to the reader, we have now additionally added the information regarding possible CRPs more specifically to the abstract and the main text (changes highlighted in yellow).

Abstract:

Here we show that large random DNA pools with a segmented structure of alternating constant and randomly generated portions are able to calculate **distinct outputs from millions of inputs** in a specific and reproducible manner, in analogy to physical unclonable functions.

Line 212ff:

The results show that if a single base in the input primer is changed, the CUF returns a completely unrelated response. Consequently, the number of challenge response pairs is 4^n , n being the number of input bases. If the input is set to 13 bases (experiments 1-12), this results in approx. 67 million CRPs.

1. The Cost of sequencing a Human Genome. *National Human Genome Institute* (2024).

<https://www.genome.gov/about-genomics/fact-sheets/Sequencing-Human-Genome-cost>

Last accessed 13th March 2024.